# On Dataset Distillation with Two-fold Pseudo-Distribution Matching

## Abstract

The goal of dataset distillation (DD) is to learn a compact synthetic dataset that maintains comparable generalization performance with the original. Distribution matching (DM), a leading DD approach, excels in addressing model scalability. However, current methods struggle with inherent feature and distribution shifts, facing a trade-off between efficiency and effectiveness. This paper reveals that in DM, models should prioritize similar samples when the image-per-class (IPC) is low, while incorporating diverse samples as IPC increases to capture broader information. We call such a finding as *capacity matching*, and then we propose a Two-fold Pseudo-Distribution matching (namely **P**seudo-**T**rajectory matching and Pseudo-**L**abel **M**atching (PTLM)) to address feature and distribution shifting issues in DM. Specifically, we design (1) a Pseudo-Trajectory Matching (PTM) to address feature shift, and (2) Pseudo-Label Matching (PLM) to address distribution shift. Our proposal is a plug-and-play component for any DM-based method. Experimental results on multiple real-world datasets show the efficiency and effectiveness of the proposed method. The source code is available at https://anonymous.4open.science/r/PTLM.

## 1 Introduction

The rapid growth of large-scale datasets has significantly advanced deep neural networks (DNNs) (Goodfellow et al., 2014; He et al., 2016; Ioffe & Szegedy, 2015; Deng et al., 2009). However, extracting their information is time-consuming and labor-intensive (Devlin, 2018; Yao et al., 2024). Additionally, the exponential increase in neural network parameters (Touvron et al., 2023) makes training on such large datasets a resource-intensive challenge. Dataset Distillation (DD) is a technique that compresses large datasets into compact, synthetic counterparts. Models trained on these distilled datasets can achieve performance comparable to those trained on the original data, thereby mitigating significant computational and storage burdens (Zhao & Bilen, 2023; Zhao et al., 2021; Wang et al., 2018).

Despite the range of DD method proposed in the literature (Cazenavette et al., 2022; Nguyen et al., 2021; Zhang et al., 2024), many existing methods are fundamentally constrained by their prohibitive computational requirements, rendering them impractical for large-scale datasets. This issue is particularly pronounced because existing approaches (Cazenavette et al., 2022; Du et al., 2023; Guo et al., 2024; Liu et al., 2024; 2023; Zhao et al., 2021; Wang et al., 2022) heavily rely on computationally intensive bi-level optimization. This reliance often makes the distillation paradigm more complex than the original model training process (Zhang et al., 2024; Wang et al., 2022). For example, trajectory matching methods (Cazenavette et al., 2022) require maintaining both the distillation model and the expert model, requiring parameter-scale squared error calculations during each training session, and also requiring the concurrent implementation of conventional classification tasks to achieve the distillation task. This approach requires significant graphics memory and computing resources, especially when applied to large-scale or high-resolution datasets, limiting its feasibility in real-world scenarios (Liu et al., 2023; Zhao et al., 2021).

One promising category of DD methods that address the above scalability issue is based on the Distribution Matching (DM) paradigm, initially proposed by Zhao and Bilen in 2023 (Zhao & Bilen, 2023). It uses learned feature extractors (from random initialization) to map original and synthetic datasets (via noise generation or random sampling) to a shared low-dimensional feature

space, achieving distillation by aligning their feature distributions. However, such approaches have two major issues: **feature shift** and **distribution shift**. The former refers to the phenomenon that feature vectors learned from randomly initialized feature extractor may different from a well-trained model on the entire original dataset. For the latter, the synthetic data may drift away from the true category regions of the original data, thereby misleading the decision boundary of the classifier trained on the synthetic data.

In this paper, we deeply analyze the DM matching pattern and propose *Matching Capacity* (see Section 2). Capacity matching refers to the concept that, when the images-per-class (IPC) is small, the synthetic dataset should prioritize learning from samples in the original dataset that closely align with its own feature distribution (typical samples). On the other hand, as the IPC increases, the synthetic dataset should integrate a wider variety of diverse samples to capture the rich information inherent in the original dataset more comprehensively (diverse samples). The distinction between typical and diverse samples is typically made on the basis of the cosine similarity average between the features of the synthetic and real datasets. Moreover, we propose a Two-fold Pseudo-Distribution matching (namely **P**seudo-**T**rajectory matching and Pseudo-**L**abel **M**atching (PTLM)) to correct the inherent feature and distribution shifts challenges (see Section 3). Specifically, (1) **Pseudo-Trajectory Matching (PTM)** combines pre-trained models and progressively injects parameters into the feature extractor throughout the distillation process, simulating training on the real dataset. Pseudo-trajectory matching aids in mitigating feature shift by enhancing the quality of feature extraction and constraining the directions of learned feature representations. (2) **Pseudo-Label Matching (PLM)** leverages the refined predictions from the original dataset on the pre-trained model to generate pseudo-labels. Then, the distribution shift is rectified by minimizing the distance between the predictions and the pseudo-labels.

As shown in Tables 1 and 2, the experimental results show that matching capacity can improve the distillation quality of DM. Simultaneously, matching capacity is a versatile plug-and-play component adaptable to any DM-based method. Moreover, experimental results across multiple datasets underscore PTLM as an effective approach, attaining state-of-the-art performance across various IPC settings while preserving competitive distillation efficiency (see Tables 3 and 4). Additionally, in cross-architecture experiments, synthetic datasets generated by PTLM were evaluated on diverse network architectures, demonstrating negligible performance degradation (see Tables 5 and 6).

The main contributions of this work are as follows:

- We propose *Maching Capacity* and its corresponding application solutions, which serve as a plug-and-play component for any DM-based method, enhancing distillation performance while maintaining efficiency.
- We propose a Two fold Pseudo-Distribution matching namely **P**seudo-**T**rajectory matching and Pseudo-**L**abel **M**atching (PTLM), which effectively alleviated the two inherent challenges of feature and distribution shifts.
- We performed extensive experiments on multiple datasets across various settings. Experimental results demonstrate that the PTLM not only achieves optimal performance but also preserves distillation efficiency.

## 2 MATCHING CAPACITY FOR DM

We explore the matching pattern of DM (Zhao & Bilen, 2023) and uncover a notable phenomenon: the relationship between image-per-class (IPC) count and synthetic dataset composition. When IPC is small, synthetic datasets should prioritize learning from samples in the original dataset that reflect its characteristics (Samples from the original dataset exhibiting high representation proximity to the synthetic dataset in the feature space.). As IPC increases, synthetic datasets benefit from incorporating a wider variety of samples to capture the full breadth of information.

Building on this insight, we introduce *Matching Capacity*, which emphasizes a critical balance in the composition of synthetic dataset. For small synthetic datasets, including diverse samples can degrade performance, while including mainly typical samples can help any model to focus on the most common features of each category. Conversely, large synthetic datasets, with their "data-hungry" nature, have the capacity to incorporate more diverse samples, enabling the model to learn a rich set of features, hence attaining best classification performance.

Table 1: **Comparison of matching capacity with DM on Fashion-MNIST.** IPC: image(s) per class. Ratio (%): the ratio of condensed examples to the whole training set. Best results are in red and the second best results are in **bold**.

| Dataset | Fashion-MNIST | | | | | | | |
|---|---|---|---|---|---|---|---|---|
| IPC | 1 | 10 | 50 | 100 | 200 | 300 | 500 | 1000 |
| Ratio | 0.02 | 0.17 | 0.83 | 1.67 | 3.33 | 5.00 | 8.33 | 16.67 |
| DM | $70.7_{\pm0.6}$ | $83.5_{\pm0.3}$ | $88.1_{\pm0.6}$ | $\mathbf{89.7_{\pm0.4}}$ | $\mathbf{91.2_{\pm0.8}}$ | $91.5_{\pm0.7}$ | $\mathbf{92.2_{\pm0.8}}$ | $\mathbf{92.6_{\pm0.4}}$ |
| Typical Samples | $\mathbf{71.1_{\pm0.8}}$ | $\mathbf{85.1_{\pm0.7}}$ | $\mathbf{88.7_{\pm0.3}}$ | $89.2_{\pm0.6}$ | $90.9_{\pm0.2}$ | $91.0_{\pm0.7}$ | $91.8_{\pm0.9}$ | $92.1_{\pm0.8}$ |
| Diverse Samples | $70.1_{\pm0.3}$ | $81.4_{\pm0.7}$ | $\mathbf{88.5_{\pm0.3}}$ | $89.9_{\pm0.6}$ | $91.6_{\pm0.8}$ | $92.1_{\pm0.5}$ | $92.7_{\pm0.6}$ | $93.2_{\pm0.1}$ |
| Whole Dataset | $93.5_{\pm0.1}$ | | | | | | | |

Table 2: **Comparison of matching capacity with DM on CIFAR-10.** IPC: image(s) per class. Ratio (%): the ratio of condensed examples to the whole training set. Best results are in red and the second best results are in **bold**.

| Dataset | CIFAR-10 | | | | | | | |
|---|---|---|---|---|---|---|---|---|
| IPC | 1 | 10 | 50 | 100 | 200 | 300 | 500 | 1000 |
| Ratio | 0.02 | 0.2 | 1 | 2 | 4 | 6 | 10 | 20 |
| DM | $26.0_{\pm0.8}$ | $48.9_{\pm0.6}$ | $63.0_{\pm0.4}$ | $67.8_{\pm0.4}$ | $72.4_{\pm0.6}$ | $73.5_{\pm0.8}$ | $75.5_{\pm0.6}$ | $76.8_{\pm0.5}$ |
| Similar Samples | $27.2_{\pm0.9}$ | $50.4_{\pm0.6}$ | $62.8_{\pm0.8}$ | $67.8_{\pm0.4}$ | $71.5_{\pm0.7}$ | $73.4_{\pm0.5}$ | $73.9_{\pm0.1}$ | $74.3_{\pm0.2}$ |
| Diverse Samples | $25.4_{\pm1.1}$ | $48.7_{\pm0.8}$ | $63.1_{\pm0.6}$ | $67.9_{\pm0.8}$ | $72.8_{\pm0.3}$ | $73.8_{\pm0.5}$ | $76.1_{\pm0.7}$ | $77.4_{\pm0.4}$ |
| Whole Dataset | $84.8_{\pm0.1}$ | | | | | | | |

This principle aligns with everyday analogies. For instance, a hard drive with limited storage must prioritize essential files for a project, while a high-capacity drive can store a diverse array of files without compromising utility. Similarly, understanding and optimizing Matching Capacity is pivotal for enhancing the efficiency and effectiveness of synthetic dataset representation and usage.

The following content will present the implementation of the matching capacity. Formally, let $z_j^{\text{syn}}$ and $z_i^{\text{real}}$ represent the low-dimensional feature distribution of the synthetic dataset sample $x_j^{\text{syn}}$ and the original dataset sample $x_i^{\text{real}}$ after being mapped by the initial feature extractor $f(\boldsymbol{\theta}, \cdot)$ with parameter distribution $\theta$.

$$z_i^{\text{real}} = f(\boldsymbol{\theta}, x_i^{\text{real}}), \quad z_j^{\text{syn}} = f(\boldsymbol{\theta}, x_j^{\text{syn}}) \tag{1}$$

To reduce the computational cost and maintain the distillation efficiency, we use Eq. 2 to calculate the batch mean of the synthetic dataset $\mathcal{B}_{\text{syn}}$ .

$$z_{\text{mean}}^{\text{syn}} = \frac{\sum_{i=1}^{|\mathcal{B}_{\text{syn}}|} f(\boldsymbol{\theta}, x_i^{\text{syn}})}{|\mathcal{B}_{\text{syn}}|} \tag{2}$$

Subsequently, the similarity score, denoted as $s_i$, between $z_i^{\text{real}}$ and $z_{\text{mean}}^{\text{syn}}$ can be obtained using the Eq. 3.

$$s_i = \frac{z_{\text{mean}}^{\text{syn}} \cdot z_i^{\text{real}}}{\left\| z_{\text{mean}}^{\text{syn}} \right\| \left\| z_i^{\text{real}} \right\|} \tag{3}$$

The mean score $s_{mean}$ is calculated by Eq. 2 and serves as an indicator for selecting typical or diverse samples. In the distillation process, we follow the basic setting of DM and use all batch samples for training at the beginning. As the distillation process unfolds through successive iterations, we employ the iteration ratio $\frac{t_i}{|T|}$ as the pruning factor. This allows us to incrementally select suitable samples from the original dataset. The corresponding calculation formula is presented as follows:

$$x_{\text{select}}^{\text{real}} = \begin{cases} \boldsymbol{s}_i \leq s_{\max} - \dfrac{t_i}{|\boldsymbol{T}|}(\boldsymbol{s}_{max} - \boldsymbol{s}_{mean}), & \text{if IPC} \geq \alpha, \\[3mm] \boldsymbol{s}_i > \boldsymbol{s}_{\min} + \dfrac{\boldsymbol{t}_i}{|\boldsymbol{T}|}(\boldsymbol{s}_{mean} - \boldsymbol{s}_{min}), & otherwise \end{cases} \tag{4}$$

In Eq. 4, $\boldsymbol{s}_{max}$ and $\boldsymbol{s}_{min}$ represent the maximum and minimum similarity scores observed in the batch samples. The variables $\boldsymbol{t}_i$ and $\boldsymbol{T}$ represent the current iteration number and total iteration number, respectively. $\alpha$ is a IPC threshold (defaults to 50). When IPC is above the threshold, i.e., the IPC is high, the more typical samples are gradually discarded. This approach forces the synthetic dataset to learn more information about the original dataset. Conversely, when IPC is small, samples with large differences are gradually discarded to learn the common features.

As shown in Tables 1 and 2, the experimental results show that the quality of the synthetic dataset is effectively improved via matching capacity. In the Fashion-MNIST (Xiao et al., 2017), when the IPC is below 50, selecting analogous samples can result in a distillation performance that is **0.6% to 2.1%** superior to that achieved by selecting all samples in the batch. When IPC is greater than 50, the difference samples have a beneficial effect, assisting in the enhancement of the model's generalization capacity by 0.2% to 0.6%. The results obtained on the CIFAR-10 (Krizhevsky et al., 2009) dataset are comparable, with a generalization performance improvement of **1.2% - 1.8%** and **0.1% - 0.6%**, respectively. It is noteworthy that the two datasets exhibited struggle when IPC was 50 and 100. This was particularly evident in the CIFAR-10 dataset, where selecting typical or diverse samples could enhance the model's generalization performance when the IPC was 50. Consequently, future research directions will encompass the investigation of adaptive IPC judgment.

## 3 METHODOLOGY

To tackle feature and distribution shifts, we draw inspiration from DANCE (Zhang et al., 2024) and propose a Two-fold Pseudo-Distribution matching (namely **P**seudo-**T**rajectory matching and Pseudo-**L**abel **M**atching (PTLM)). As shown in Fig. 1, pseudo-trajectory matching simulates training dynamics on original datasets to guide feature representation direction, improving distillation quality when combined with capacity matching. Concurrently, pseudo-label matching reduces divergence between predicted outputs of synthetic and original datasets, correcting distribution inconsistencies. Sections 3.1 and 3.2 elaborate our proposed solutions.

### 3.1 PSEUDO-TRAJECTORY MATCHING

DM (Zhao & Bilen, 2023) maps $\boldsymbol{D}_{real}$ and $\boldsymbol{D}_{syn}$ to a low-dimensional space through random initial feature extractors and enables $\boldsymbol{D}_{syn}$ to contain rich information from the original dataset by aligning feature representations. In fact, the randomly initialized feature extractors cannot maintain the proxy reliability during the distillation process, which will cause the distribution of $\boldsymbol{D}_{syn}$ to gradually move away from $\boldsymbol{D}_{real}$. Some studies (Wang et al., 2022; Zhang et al., 2024) have noticed the above problems.

CAFE (Wang et al., 2022) fully trains multiple models and aligns features to improve feature representation. DANCE (Zhang et al., 2024) involves randomly cropping pre-trained models and fusing them with feature extractors, which is intended to enhance the representation capabilities of the initial model and improve the performance of the distillation. Nevertheless, the implementation of the CAFE necessitates a considerable computational burden. Although DANCE limits the sample representation direction through pre-trained models, random cropping may result in a negative parameter fusion, which in turn causes the extracted features to move from aggregation to a more discrete distribution. This, in turn, may lead to instability in the datasets distillation process.

To overcome feature shift, we propose a **Pseudo-trajectory matching (PTM)** to constrain the representation direction of feature extractors in samples belonging to the same class. Specifically, we introduced pre-trained models $f(\boldsymbol{\theta}_i^*, \cdot)$ and gradually injected parameters $\boldsymbol{\theta}_i^*$ into the randomized initial feature extractor $f(\boldsymbol{\theta}, \cdot)$ to simulate the neural network training on the real dataset from the parameter optimization level. By simulating the training process, the feature extractor is capable of normalizing the mapping directions of samples with similar characteristics, thus preventing feature

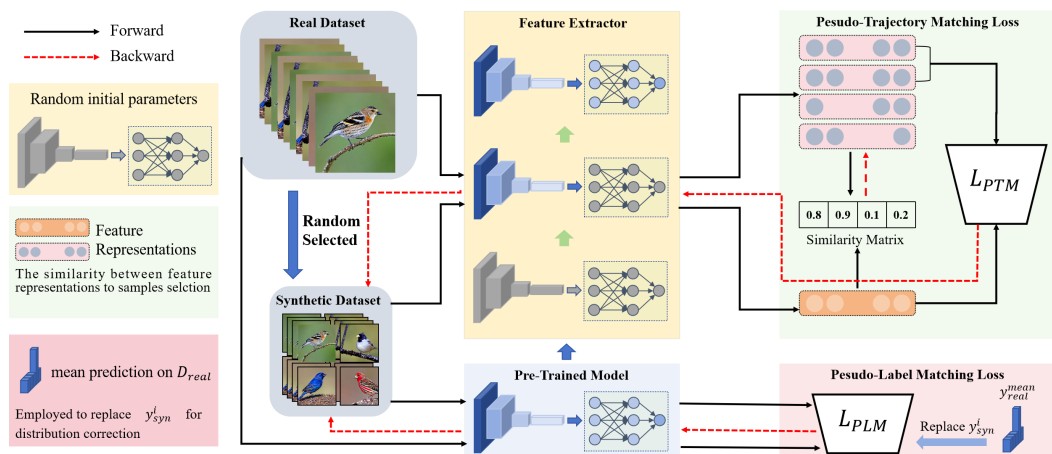

Figure 1: **The framework of PTLM**. The synthetic dataset is generated using random noise or sampling. **(1)** The yellow section shows pseudo-trajectory matching to reduce feature shifts. **(2)** Sharpened original dataset predictions align synthetic predictions to address distribution shifts. **(3)** The green section uses a similarity matrix to select training samples based on IPC. Finally, the synthetic dataset is optimized directly by minimizing the loss functions associated with both $\mathcal{L}_{\text{PTM}}$ and $\mathcal{L}_{\text{PLM}}$.

shift and maintaining the consistency of the real dataset. Formally, the process can be expressed by Eq. 5:

$$\boldsymbol{\theta}_i = \frac{t_i/\lambda}{|T|} \cdot \boldsymbol{\theta}_i^* + (1 - \frac{t_i/\lambda}{|T|}) \cdot \boldsymbol{\theta}_i \tag{5}$$

We employ the ratio of the current distillation step $t_i$ to the total number of distillation steps $|T|$ as the injection ratio. Researches (Guo et al., 2024; Arpit et al., 2017; Zhang et al., 2021) suggest that synthetic datasets with varying cardinalities are best suited to different stages of pre-trained model training. Consequently, we introduce an additional scaling factor $\lambda$ to fine-tune the injection ratio, facilitating the alignment of training trajectories across different phases. As the distillation progresses, the extractor parameters will gradually approach the pre-trained model. Then the feature alignment loss is calculated by Eq 6:

$$\mathcal{L}_{\text{PTM}} = \left\| \frac{\sum_{i=1}^{|\mathcal{D}_{\text{real}}|} f(\boldsymbol{\theta}_i, \boldsymbol{x}_i^{\text{real}})}{|\mathcal{D}_{\text{real}}|} - \frac{\sum_{j=1}^{|\mathcal{D}_{\text{syn}}|} f(\boldsymbol{\theta}_i, \boldsymbol{x}_i^{\text{syn}})}{|\mathcal{D}_{\text{syn}}|} \right\|^2. \tag{6}$$

In contrast to the continuous parameter updates employed by the CAFE (Wang et al., 2022), the introduction of pre-trained models enhances the feature alignment capability without the need for any hyperparameters, thereby significantly reducing the computational cost. Compared to DANCE (Zhang et al., 2024), the pressure of feature shift is alleviated by the random cropping and fusion of pre-trained models. However, the crop ratio of the models is uncontrollable, which may result in reverse distillation. Pseudo-trajectory matching serves to guarantee that feature extractors are consistently progressing in the optimal direction by simulating the real dataset training. Through the utilization of adaptive inject parameters weights, it is possible to emulate the training process on the real dataset, thereby ensuring the quality of feature alignment. It is worth mentioning that this module can combine the concept of matching capacity, as previously proposed, to further enhance the quality of the synthetic dataset after distillation.

## 3.2 PSEUDO-LABEL MATCHING

Distribution shift poses a significant challenge to all DM-based distillation methods. Neglecting the inter-class distribution constraints can lead to sample overlap or misplacement within synthetic datasets. This, in turn, impedes the establishment of effective decision boundaries for classifiers. Moreover, the severity of this issue escalates as the number of categories increases. Prior research (Zhang et al., 2024) shows that while pre-trained models accurately classify original dataset, they struggle with synthetic datasets, further highlighting the problem and compromising model generalization.

To address this, we introduce **Pseudo-Label Matching (PLM)**, a novel method to mitigate distribution shifts in synthetic datasets. Inspired by traditional image classification principles (Wang et al., 2017; Akata et al., 2013; Maurício et al., 2023; Wang et al., 2023), which rely on cross-entropy loss to iteratively refine model parameters, PLM adopts a complementary approach. Instead of adjusting parameters with fixed inputs, PLM fixes neural network parameters and dynamically refines input data. This enables approximate pseudo-label predictions for modified inputs, aligning synthetic data distributions with target datasets.

Specifically, the Pseudo-Labeling Matching(PLM) is an auxiliary classification task that incorporates pseudo-labeling techniques, effectively correcting distribution shift and serving as a regularization mechanism. During implementation, we first freeze the parameters of the pre-trained model, generate predictions for mini-batch samples of each class in the real dataset, sharpen the predictions, and compute their class-wise average to use as pseudo-labels for corresponding classes in the synthetic dataset. We then align the predictive distributions of the synthetic and real datasets using conventional classification optimization methods, and directly apply the aligned results to the synthetic dataset to achieve distribution correction.

Building on this approach, we integrate a classification task into the pre-trained model to rectify the distribution. To achieve closer alignment with the real dataset's distribution, we generate a soft and pseudo-label based on the predictions from the real dataset, replacing the original labels and thereby strengthening the correction process. Specifically, $x_j^{\text{syn}}$ and $x_i^{\text{real}}$ are input into the pre-trained model $f(\boldsymbol{\theta}^*, \cdot)$ to obtain the prediction $\boldsymbol{y}_j^{\text{syn}}$ and $\boldsymbol{y}_i^{\text{real}}$, both being a distribution over all the categories. To enhance the quality of pseudo labels, the samples that cannot be correctly classified in $\boldsymbol{y}_i^{\text{real}}$ are filtered out, since the real dataset may have noise labels or abnormal samples (Tanaka et al., 2018; Zhang & Sabuncu, 2018; Wei et al., 2020; Yu et al., 2019; Tan et al., 2021). Then Eq. 7 calculates the average distribution of original data for a category $c$.

$$y_{\text{avg}}^{real} = \frac{\sum_{i=1}^{|\mathcal{B}_{\text{real}}^{\text{true}}|} \{(\boldsymbol{y}_i^{\text{real}} \mid \max_j \boldsymbol{y}_i^{\text{real}} = c\}}{|\mathcal{B}_{\text{real}}^{\text{true}}|} \tag{7}$$

We further introduce a sharpening factor $\tau$ to scale the $y_{\text{avg}}^{real}$ in Eq 8, which is uniformly set to 0.1 in all experiments. We then use the probability distribution $\hat{\boldsymbol{y}}^{\text{real}}$ as the pseudo-labels, substituting the original labels for the supplementary classification task.

$$\hat{\boldsymbol{y}}^{\text{real}} = softmax(\frac{y_{\text{avg}}^{real}}{\tau}) \tag{8}$$

The generated pseudo-labels encompass more information about the original dataset's distribution than the initial labels. Unlike the original labels, the pseudo-labels derived from the original dataset's predictions help guide the synthetic dataset to more closely align with the class distribution of the original dataset. Then, the standard cross-entropy loss is employed to optimize the additional classification task:

$$\mathcal{L}_{\text{PLM}} = -\frac{1}{|\boldsymbol{D}_{\text{syn}}|} \sum_{i=1}^{|\boldsymbol{D}_{\text{syn}}|} l(y_i^{\text{syn}}, \hat{\boldsymbol{y}}) \tag{9}$$

Finally, we integrate the $\mathcal{L}_{\text{PTM}}$ of pseudo-trajectory matching and the $\mathcal{L}_{\text{PLM}}$ of pseudo-label matching into a unified framework to obtain the final loss function:

$$\mathcal{L}_{\text{PTLM}} = \mathcal{L}_{\text{PTM}} + \mathcal{L}_{\text{PLM}} \tag{10}$$

Distinguishing itself from prior research (Zhao & Bilen, 2023; Wang et al., 2022; Zhao et al., 2023; Zhang et al., 2024), our approach achieves pseudo-trajectory matching through controlled parameter injection, effectively alleviating feature shifts without introducing additional hyperparameters or requiring extra training steps. Building on foundational principles of traditional image classification tasks, we propose a pseudo-label matching strategy designed to address inter-class distribution imbalances within the synthetic dataset. This method aligns the class distribution of the synthetic dataset more closely with that of the original dataset, thereby enhancing its representational fidelity and ensuring a more robust learning process.

## 4 EXPERIMENTS

To evaluate PTLM, we developed a comprehensive experimental system across varying data resolutions. Then an ablation study confirms the reliability of the proposed modules. Owing to space limitations, We provide the analysis of efficiency and ablation study in Appendix sections C and section D. Additionally, sensitivity analysis of the hyperparameters is provided in Appendix section E, and visualization of the syntheticsized datasets are provided in Appendix section F.

### 4.1 EXPERIMENTAL SETUP

**Datasets and Network Structure.** To rigorously assess the feasibility and robustness of the proposed PTLM framework, we carried out an extensive series of experiments leveraging a diverse range of datasets encompassing low-, mid-, and high-resolution categories. Specifically, Fashion-MNIST (Xiao et al., 2017) and CIFAR-10/100 (Krizhevsky et al., 2009) are used as low-resolution datasets, while TinyImageNet (Le & Yang, 2015) and subsets of ImageNet (Deng et al., 2009) are used as mid- and high-resolution datasets, respectively. To better align with practical applications, a three-layer convolutional neural network (CNNs) (LeCun et al., 1998) is employed as the distilled model for low-resolution datasets, with an additional layer added for medium- and high-resolution datasets.

**Evaluation Metrics.** Following previous researches (Zhao & Bilen, 2023; Zhang et al., 2024), the test accuracy of the network trained on the distillation dataset $\boldsymbol{D}_{syn}$ is used as the main metric. To eliminate randomness, we train 10 times on the low- and mid-resolution datasets, and 5 times on the high-resolution datasets. The mean test accuracy is employed as the evaluation value, with the standard deviation duly reported. The evaluation of distillation efficiency is based on the computation time and the usage of GPU memory per step. The computation time is the average value of every 1,000 iterations.

**Implementation Details.** For training pre-trained models, we use the SGD optimizer (learning rate: 0.01, momentum: 0.9, weight decay: 0.0005). Models are trained for 70 epochs on low-resolution datasets (Fashion-MNIST (Xiao et al., 2017), CIFAR-10/100 (Krizhevsky et al., 2009)), mid-resolution datasets (TinyImageNet (Le & Yang, 2015)), and 100 epochs on high-resolution datasets (six ImageNet subsets (Deng et al., 2009)). Five pre-trained models are prepared and randomly selected during distillation. For pseudo-trajectory and pseudo-label matching, the SGD optimizer uses a learning rate of 0.1 for ImageNet subsets and 0.01 for other datasets, scaled by IPC. Data augmentation (Zhao & Bilen, 2021) (e.g., color transformation, cropping, CutMix (Yun et al., 2019)) is applied with factor parameters of 2 (low-/mid-resolution datasets) and 3 (ImageNet subsets (Kim et al., 2022)). Synthetic datasets are initialized via random sampling to accelerate optimization. Experiments run on two NVIDIA Tesla A30 GPUs and an A100 GPU, with distillation evaluation exclusively on the A100.

**Baselines.** For a comprehensive comparison, we identified and selected three categories of methods to serve as baselines. The core selection-based methods include Random Selection, Herding (Welling, 2009), and K-center (Farahani & Hekmatfar, 2009). DC (Zhao et al., 2021), DSA (Zhao & Bilen, 2021), IDC (Kim et al., 2022), Dream (Liu et al., 2023) and MTT (Cazenavette et al., 2022) are included in the two-level optimization-based methods. The third category is the

DM-based methods including CAFE (Wang et al., 2022), DM (Zhao & Bilen, 2023), IDM (Zhao et al., 2023), and DANCE (Zhang et al., 2024).

Table 3: **Comparison with previous coreset selection and dataset distillation methods on low-resolution datasets and medium-resolution datasets.** IPC: image(s) per class. Ratio (%): the ratio of distillation examples to the whole training set. Best results are **highlighted** and the second best results are in **bold**. Note that some entries are marked as "-" because of scalability issues or the results are not reported.

| Dataset | Fashion-MNIST | | | CIFAR-10 | | | CIFAR-100 | | | TinyImageNet | | |
|---|---|---|---|---|---|---|---|---|---|---|---|---|
| IPC | 1 | 10 | 50 | 1 | 10 | 50 | 1 | 10 | 50 | 1 | 10 | 50 |
| Ratio | 0.02 | 0.17 | 0.83 | 0.02 | 0.2 | 1 | 0.02 | 0.2 | 1 | 0.2 | 2 | 10 |
| Random | $51.4_{\pm3.8}$ | $73.8_{\pm0.7}$ | $82.5_{\pm0.7}$ | $14.4_{\pm2.0}$ | $26.0_{\pm1.2}$ | $43.4_{\pm1.0}$ | $4.2_{\pm0.3}$ | $14.6_{\pm0.5}$ | $30.0_{\pm0.4}$ | $1.4_{\pm0.1}$ | $5.0_{\pm0.2}$ | $15.0_{\pm0.4}$ |
| Herding | $67.0_{\pm1.9}$ | $71.1_{\pm0.7}$ | $71.9_{\pm0.8}$ | $21.5_{\pm1.2}$ | $31.6_{\pm0.7}$ | $40.4_{\pm0.6}$ | $8.4_{\pm0.3}$ | $17.3_{\pm0.3}$ | $33.7_{\pm0.5}$ | $2.8_{\pm0.2}$ | $6.3_{\pm0.2}$ | $16.7_{\pm0.3}$ |
| K-Center | $66.9_{\pm1.8}$ | $54.7_{\pm1.5}$ | $68.3_{\pm0.8}$ | $21.5_{\pm1.3}$ | $14.7_{\pm0.9}$ | $27.0_{\pm1.4}$ | $8.3_{\pm0.3}$ | $7.1_{\pm0.2}$ | $30.5_{\pm0.3}$ | - | - | - |
| DC | $70.5_{\pm0.6}$ | $82.3_{\pm0.4}$ | $83.6_{\pm0.4}$ | $28.3_{\pm0.5}$ | $44.9_{\pm0.5}$ | $53.9_{\pm0.5}$ | $12.8_{\pm0.3}$ | $25.2_{\pm0.3}$ | - | $5.3_{\pm0.1}$ | $12.9_{\pm0.1}$ | $12.7_{\pm0.4}$ |
| DSA | $70.6_{\pm0.6}$ | $84.6_{\pm0.3}$ | **$88.7_{\pm0.2}$** | $28.8_{\pm0.7}$ | $52.1_{\pm0.5}$ | $60.6_{\pm0.5}$ | $13.9_{\pm0.3}$ | $32.3_{\pm0.3}$ | $42.8_{\pm0.4}$ | $5.7_{\pm0.1}$ | $16.3_{\pm0.2}$ | $5.1_{\pm0.2}$ |
| IDC | $81.0_{\pm0.2}$ | $86.0_{\pm0.3}$ | $86.2_{\pm0.2}$ | $50.6_{\pm0.4}$ | $67.5_{\pm0.5}$ | $74.5_{\pm0.1}$ | - | $45.1_{\pm0.4}$ | - | - | - | - |
| DREAM | $81.3_{\pm0.2}$ | $86.4_{\pm0.3}$ | $86.8_{\pm0.3}$ | **$51.1_{\pm0.3}$** | $69.4_{\pm0.4}$ | $74.8_{\pm0.1}$ | **$29.5_{\pm0.3}$** | $46.8_{\pm0.7}$ | $52.6_{\pm0.4}$ | $10.0_{\pm0.4}$ | - | **$29.5_{\pm0.3}$** |
| MTT | - | - | - | $31.9_{\pm1.2}$ | $56.4_{\pm0.7}$ | $65.9_{\pm0.6}$ | $24.3_{\pm0.3}$ | $40.1_{\pm0.4}$ | $47.7_{\pm0.2}$ | $6.2_{\pm0.4}$ | $17.3_{\pm0.2}$ | $26.5_{\pm0.3}$ |
| CAFE | $77.1_{\pm0.9}$ | $83.0_{\pm0.4}$ | $84.8_{\pm0.4}$ | $30.3_{\pm1.1}$ | $46.3_{\pm0.6}$ | $55.5_{\pm0.6}$ | $12.9_{\pm0.3}$ | $27.8_{\pm0.3}$ | $37.9_{\pm0.3}$ | - | - | - |
| CAFE+DSA | $73.7_{\pm0.7}$ | $83.0_{\pm0.3}$ | **$88.2_{\pm0.3}$** | $31.6_{\pm0.8}$ | $50.9_{\pm0.5}$ | $62.3_{\pm0.4}$ | $14.0_{\pm0.3}$ | $31.5_{\pm0.2}$ | $42.9_{\pm0.2}$ | - | - | - |
| DM | $70.7_{\pm0.6}$ | $83.5_{\pm0.3}$ | $88.1_{\pm0.6}$ | $26.0_{\pm0.8}$ | $48.9_{\pm0.6}$ | $63.0_{\pm0.4}$ | $11.4_{\pm0.3}$ | $29.7_{\pm0.3}$ | $43.6_{\pm0.4}$ | $3.9_{\pm0.2}$ | $12.9_{\pm0.4}$ | $24.1_{\pm0.3}$ |
| IDM | - | - | - | $45.6_{\pm0.7}$ | $58.6_{\pm0.1}$ | $67.5_{\pm0.1}$ | $20.1_{\pm0.3}$ | $45.1_{\pm0.1}$ | $50.0_{\pm0.2}$ | $10.1_{\pm0.2}$ | $21.9_{\pm0.2}$ | $27.7_{\pm0.3}$ |
| DataDAM | - | - | - | $32.0_{\pm1.2}$ | $54.2_{\pm0.8}$ | $67.0_{\pm0.4}$ | $14.5_{\pm0.5}$ | $34.8_{\pm0.5}$ | $49.4_{\pm0.3}$ | $8.3_{\pm0.4}$ | $18.7_{\pm0.3}$ | $28.7_{\pm0.3}$ |
| DANCE | **$81.5_{\pm0.4}$** | $86.3_{\pm0.2}$ | $86.9_{\pm0.1}$ | $47.1_{\pm0.2}$ | **$70.8_{\pm0.2}$** | **$76.1_{\pm0.1}$** | $27.9_{\pm0.2}$ | **$49.8_{\pm0.1}$** | $52.8_{\pm0.1}$ | **$11.6_{\pm0.2}$** | **$26.4_{\pm0.3}$** | $28.9_{\pm0.4}$ |
| **PTLM** (Ours) | **$82.5_{\pm0.3}$** | **$87.3_{\pm0.2}$** | **$88.7_{\pm0.2}$** | $48.9_{\pm0.3}$ | **$71.3_{\pm0.3}$** | **$78.1_{\pm0.2}$** | **$29.5_{\pm0.3}$** | **$51.2_{\pm0.2}$** | **$55.1_{\pm0.3}$** | **$13.3_{\pm0.3}$** | **$27.1_{\pm0.4}$** | **$29.7_{\pm0.2}$** |
| Whole Dataset | $93.5_{\pm0.1}$ | | | $84.8_{\pm0.1}$ | | | $56.2_{\pm0.3}$ | | | $37.6_{\pm0.4}$ | | |

Table 4: **Comparison with previous coreset selection and dataset distillation methods on high-resolution** ($128 \times 128$) **Imagenet-Subsets.** All the datasets are distillation using a 5-layer ConvNet.

| Dataset | ImageNette | | ImageWoof | | ImageFruit | | ImageMeow | | ImageSquawk | | ImageYellow | |
|---|---|---|---|---|---|---|---|---|---|---|---|---|
| IPC | 1 | 10 | 1 | 10 | 1 | 10 | 1 | 10 | 1 | 10 | 1 | 10 |
| Ratio | 0.105 | 1.050 | 0.110 | 1.100 | 0.077 | 0.77 | 0.077 | 0.77 | 0.077 | 0.77 | 0.077 | 0.77 |
| Random | $23.5_{\pm4.8}$ | $47.7_{\pm2.4}$ | $14.2_{\pm0.9}$ | $27.0_{\pm1.9}$ | $13.2_{\pm0.8}$ | $21.4_{\pm1.2}$ | $13.8_{\pm0.6}$ | $29.0_{\pm1.1}$ | $21.8_{\pm0.5}$ | $40.2_{\pm0.4}$ | $20.4_{\pm0.6}$ | $37.4_{\pm0.5}$ |
| MTT | $47.7_{\pm0.9}$ | $63.0_{\pm1.3}$ | $28.6_{\pm0.8}$ | $35.8_{\pm1.8}$ | $26.6_{\pm0.8}$ | $40.3_{\pm1.3}$ | $30.7_{\pm1.6}$ | $40.4_{\pm2.2}$ | $39.4_{\pm1.5}$ | $52.3_{\pm1.0}$ | $45.2_{\pm0.8}$ | $60.0_{\pm1.5}$ |
| DM | $32.8_{\pm0.5}$ | $58.1_{\pm0.3}$ | $21.1_{\pm1.2}$ | $31.4_{\pm0.5}$ | - | - | - | - | $31.2_{\pm0.7}$ | $50.4_{\pm1.2}$ | - | - |
| DataDAM | $34.7_{\pm0.9}$ | $59.4_{\pm0.4}$ | $24.2_{\pm0.5}$ | $34.4_{\pm0.4}$ | - | - | - | - | $36.4_{\pm0.8}$ | $55.4_{\pm0.9}$ | - | - |
| DANCE | **$57.2_{\pm0.5}$** | **$80.2_{\pm0.7}$** | **$30.6_{\pm0.3}$** | **$57.8_{\pm1.1}$** | **$30.6_{\pm0.8}$** | **$52.8_{\pm0.7}$** | **$39.4_{\pm0.8}$** | **$60.4_{\pm1.1}$** | **$52.0_{\pm0.5}$** | **$77.2_{\pm0.3}$** | **$51.8_{\pm1.1}$** | **$78.8_{\pm0.7}$** |
| **PTLM** (Ours) | **$60.3_{\pm0.8}$** | **$82.3_{\pm0.9}$** | **$34.5_{\pm1.2}$** | **$61.2_{\pm1.1}$** | **$32.1_{\pm0.8}$** | **$57.8_{\pm0.4}$** | **$40.7_{\pm0.4}$** | **$62.1_{\pm0.7}$** | **$55.8_{\pm1.0}$** | **$80.7_{\pm1.0}$** | **$53.7_{\pm0.9}$** | **$81.0_{\pm1.2}$** |
| Whole Dataset | $87.4_{\pm1.0}$ | | $67.0_{\pm1.3}$ | | $63.9_{\pm2.0}$ | | $66.7_{\pm1.1}$ | | $87.5_{\pm0.3}$ | | $84.4_{\pm0.6}$ | |

## 4.2 COMPARATIVE EXPERIMENTS EVALUATION

**Low-resolution Datasets.** To evaluate distillation performance on low-dimensional datasets, we tested PTLM using Fashion-MNIST, CIFAR-10, and CIFAR-100. The results in Table 3 show that PTLM outperforms current SOTA methods by **1.0%-2.3%** across various IPC settings. For CIFAR-10, PTLM achieved the best results in all settings except IPC 1. Notably, at IPC 10 and 50, it surpassed the next-best model by **0.5%** and **2.0%**, respectively. In CIFAR-100, PTLM consistently led across all settings, particularly at IPC 50, where it exceeded the suboptimal method by **2.3%**, reaching an accuracy of 55.1%. This is just 1.1% below the 56.2% accuracy of the full dataset, despite using only 1% of the data. However, in the IPC 1 scenario, PTLM's performance was only comparable to the baseline, likely due to the insufficient information contained in a single image when dealing with larger datasets.

**Mid-resolution Datasets.** We used TinyImageNet to evaluate PTLM's distillation performance on medium-resolution datasets. The results in Table 3 demonstrate PTLM's superior performance compared to current SOTA methods across all settings. When IPC is 10, PTLM outperforms by **0.7%**, with the margin increasing to **0.8%** and **1.7%** at IPC 50 and 1, respectively. Unlike the suboptimal performance observed for low-resolution datasets at IPC 1, this phenomenon does not occur in the medium-resolution setting. This is due to TinyImageNet's smaller dataset size—500 samples per class compared to 5,000 in CIFAR-10—which reduces the real dataset's informational richness, making each image represent a larger proportion of the dataset's content. Additionally,

TinyImageNet's resolution of $64 \times 64$ (three times larger than CIFAR's $32 \times 32$) enhances the ability to extract information, aligning with the concept of matching capacity. As IPC or resolution increases, the distilled dataset's capacity to capture and represent information expands accordingly.

**High-resolution Datasets.** The distillation performance on high-resolution datasets holds greater importance compared to low- and medium-resolution datasets, as it more accurately reflects real-world conditions. Specifically, we selected six subsets from ImageNet-1k (Deng et al., 2009) as the distillation targets. For ease of processing, the images in the dataset were uniformly cropped resolution to $128 \times 128$. The experimental results are presented in Table 4, where PTLM demonstrates the highest distillation performance across all evaluated datasets. It is worth noting that we achieved a test accuracy of more than **80%** in ImageNette, ImageSquawk, and ImageYellow using only **10%** of the samples in the original dataset. Although ImageWoof, ImageFruit, and Image-meow are challenging datasets, PTLM is still able to achieve distillation performance surpassing the current state-of-the-art technology.

Table 5: **Cross-architecture generalization performance** (%) **on CIFAR-10.** The synthetic data is condensed using ConvNet-3 and evaluated using other architectures. The best results are in **red**.

| Method | IPC | ConvNet-3 | ResNet-10 | DenseNet-121 |
|---|---|---|---|---|
| DSA | 10 | $52.1_{\pm 0.5}$ | $32.9_{\pm 0.3}$ | $34.5_{\pm 0.1}$ |
| | 50 | $60.6_{\pm 0.5}$ | $49.7_{\pm 0.4}$ | $49.1_{\pm 0.2}$ |
| IDC | 10 | $67.5_{\pm 0.5}$ | $63.5_{\pm 0.1}$ | $61.6_{\pm 0.6}$ |
| | 50 | $74.5_{\pm 0.1}$ | $72.4_{\pm 0.5}$ | $\mathbf{71.8}_{\pm 0.6}$ |
| MTT | 10 | $56.4_{\pm 0.7}$ | $34.5_{\pm 0.8}$ | $41.5_{\pm 0.5}$ |
| | 50 | $65.9_{\pm 0.6}$ | $43.2_{\pm 0.4}$ | $51.9_{\pm 0.3}$ |
| DM | 10 | $48.9_{\pm 0.6}$ | $42.3_{\pm 0.5}$ | $39.0_{\pm 0.1}$ |
| | 50 | $63.0_{\pm 0.4}$ | $58.6_{\pm 0.3}$ | $57.4_{\pm 0.3}$ |
| DANCE | 10 | $70.8_{\pm 0.2}$ | $67.0_{\pm 0.2}$ | $64.5_{\pm 0.3}$ |
| | 50 | $76.1_{\pm 0.1}$ | $68.0_{\pm 0.1}$ | $64.8_{\pm 0.3}$ |
| **PTLM** (Ours) | 10 | $\mathbf{71.3}_{\pm 0.3}$ | $\mathbf{67.6}_{\pm 0.3}$ | $\mathbf{64.9}_{\pm 0.3}$ |
| | 50 | $\mathbf{78.1}_{\pm 0.2}$ | $\mathbf{75.4}_{\pm 0.4}$ | $69.8_{\pm 0.2}$ |

## 4.3 CROSS-ARCHITECTURE EVALUATION

To evaluate the transferability of our distilled dataset, we conducted ten rounds of experiments across a range of diverse network architectures, using average test accuracy as the evaluation metric, as illustrated in Table 5. The experimental results indicate that PTLM not only delivers robust performance on the specific architecture used during distillation but also exhibits remarkable portability and exceptional generalization across diverse architectural configurations. This versatility underscores its significant practical value, as it can be effectively deployed in various computational environments without requiring extensive re-training or fine-tuning.

## 5 CONCLUSION

This paper presents a comprehensive analysis of the Distribution Matching (DM) paradigm, introducing the concept of **Matching Capacity** and its practical implementation. A key challenge in DM is mitigating feature and distribution shifts. To address this, we propose a Two-fold Pseudo-Distribution Matching strategy, termed (namely **P**seudo-**T**rajectory matching and Pseudo-**L**abel **M**atching (PTLM)). Matching Capacity refers to the principle of prioritizing typical samples under smaller Images Per Class (IPC) to reinforce shared features, while focusing on more diverse and possibly subtle samples under larger IPC to extract deeper insights. The PTLM framework incorporates two main components: (1) Pseudo-Trajectory Matching, which enhances distillation by simulating original training dynamics and incorporating pre-trained parameters to constrain feature shifts; and (2) Pseudo-Label Matching, which aligns predictions between synthetic and original datasets to reduce distribution shifts. Experimental results demonstrate that PTLM achieves state-of-the-art performance in knowledge distillation with high efficiency, showing strong adaptability across architectures and minimal performance degradation.

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

## A    RELATED WORK

Dataset distillation has emerged as a pivotal domain in data science, seeking to alleviate the challenges posed by data redundancy through the strategic compression of original datasets. This section provides a comprehensive chronological review of recent advancements in the field, emphasizing their strengths, limitations, and overall contributions. By synthesizing these developments, it offers a clear and insightful overview of the progress achieved and the current landscape of dataset distillation research.

### A.1    CORESET SELECTION

CS-based method is a data reduction technique that aims to select a subset of samples from the original dataset under constraints to eliminate redundant data. Herding (Welling, 2009) selects samples positioned centrally within each class, while the K-center (Farahani & Hekmatfar, 2009) identifies multiple central points across each class. FastCore (Chai et al., 2023) partitions the entire training set into multiple clusters, where each cluster comprises samples with similar feature distances (measured via Euclidean distance). By constraining the overall gradient with the maximum feature distance between clusters, FastCore enables more efficient coreset selection through iterative processing of these clusters. This approach accomplishes core-set selection without requiring machine learning model training.

### A.2    DATASETS COMPRESSION

Wang et al. (Wang et al., 2018) introduced the concept of dataset distillation, in which network parameters are represented as functions of synthetic data, and the synthetic dataset is optimized to minimize the training loss on the real data. Additionally, they incorporated alternative objectives to streamline the dataset distillation optimization process further. Gradient Matching (Zhao et al., 2021) generates the synthetic dataset by matching the weight gradients of neural networks on real and extracted training datasets. DSA (Zhao & Bilen, 2021) introduces a differentiable Siamese augmentation approach, applying identical, randomly sampled data transformations to both real and synthetic data samples in each training iteration. This technique further facilitates the backpropagation of loss function gradients for synthetic data via differentiable data transformations, enhancing optimization efficiency. MTT (Cazenavette et al., 2022) directly emulates the long-term training dynamics of a network trained on a real dataset by aligning parameter trajectory segments trained on synthetic data with pre-recorded trajectory segments from the real dataset. This method mitigates the pitfalls of myopic optimization and alleviates the challenges of modeling the entire trajectory. Based on MTT, Guo et al. (Guo et al., 2024) reveal the learning rules under different IPC settings and propose a pseudo-labeling strategy to optimize the training process. DM (Zhao & Bilen, 2023) minimizes the representation distance between random models on synthetic and real datasets to achieve

the dataset distillation. Based on DM, DANCE (Zhang et al., 2024) uses expert models to randomly match model parameters and employs an additional task to improve performance.

### A.3 Limitation of Dataset Distillation

Coreset Selection, an early paradigm in dataset distillation, primarily relies on greedy algorithms. However, these methods impose performance limitations and struggle to scale effectively with large datasets. Although dataset compression techniques demonstrate strong performance, they often rely on bi-level optimization, which incurs significant computational costs that hinder scalability. Distribution matching, a more recent distillation paradigm, offers efficient distillation capabilities but faces challenges related to distribution shifts. This occurs because random feature extractors cannot consistently serve as reliable proxies. Furthermore, as distillation progresses, interactions between samples from different classes can inadvertently distort the model's decision boundaries.

This paper begins by exploring the foundational principles of distribution matching (DM) and introduces the concept of **Capacity Matching**. It then develops strategies for **Pseudo-Trajectory Matching** and **Pseudo-Label Matching (PTLM)** to address these challenges, advancing the efficiency and robustness of the distillation process.

## B Distillation Algorithm

The pseudo-code of PTLM is described in Algorithm 1. In addition to pseudo-trajectory matching and pseudo-label matching, the popular data augmentation technique of "down- and up-sampling" is also integrated into our method. This augmentation technique involves down-sampling multiple images by a specified factor, combining them into a single composite image while preserving the reduced size for subsequent distillation. During the model inference stage, each sub-image is up-sampled to its original dimensions to enhance accuracy. This enhancement strategy was originally introduced by IDC (Kim et al., 2022) and has been widely used in various dataset distillation works (Zhang et al., 2024).

---

**Algorithm 1** Pseudo-Trajectory matching and Pseudo-Label matching (PTLM)

---

**Input**: Real training set $\mathcal{D}_{\text{real}}$
**Parameter**: Number of pre-trained models $N$; Number of current distillation iterations $t_i$; Number of distillation iterations $T$; Learning rate of the synthetic set $\eta$;
**Output**: The distilled set $\mathcal{D}_{\text{syn}}$

1: Initialize $\mathcal{D}_{\text{syn}}$ with randomly selected real data
2: Pre-train $N$ models $\{f(\boldsymbol{\theta}_i^*, \cdot)\}_{i=1}^N$ and save their corresponding initial encoders $\{f(\boldsymbol{\theta}_i, \cdot)\}_{i=1}^N$
3: **for** $t = 1, 2, \ldots, T$ **do**
4:      Randomly select a pre-trained model $\{f(\boldsymbol{\theta}_i^*, \cdot)\}$ and calculate the injection ratio $\frac{t}{|T|}$
5:      update the feature extractor parameters $\boldsymbol{\theta}_i$ by Eq. (5)
6:      Selecting the suitable samples by Eq. (4)
7:      Calculate the matching loss $\mathcal{L}_{\text{PTM}}$ by Eq. (9)
8:      Update the $\mathcal{D}_{\text{syn}}$ by $\mathcal{D}_{\text{syn}} = \mathcal{D}_{\text{syn}} - \eta \nabla_{\mathcal{D}_{\text{syn}}} \mathcal{L}_{\text{PTM}}$
9:      Calculate the classification loss $\mathcal{L}_{\text{PLM}}$ by Eq. (10)
10:      Update the $\mathcal{D}_{\text{syn}}$ by $\mathcal{D}_{\text{syn}} = \mathcal{D}_{\text{syn}} - \eta \nabla_{\mathcal{D}_{\text{syn}}} \mathcal{L}_{\text{PLM}}$
11: **end for**
12: **Return**: $\mathcal{D}_{\text{syn}}$

---

## C Training Efficiency Evaluation

In dataset distillation, computational resources and time costs are crucial factors. Based on the benchmarks from prior studies (Cazenavette et al., 2022; Zhao et al., 2021; Zhang et al., 2024), we evaluated the distillation efficiency of PTLM, focusing on computational time and GPU memory usage. Existing methods cut costs significantly compared to processing full datasets. However, high - performance distillation methods often face a trade - off: better performance comes with more

Table 6: **Time and GPU memory cost comparison of SOTA datasets condensation methods.** Run Time: the time for a single iteration. GPU memory: the peak memory usage during condensing. Both run time and GPU memory are averaged over 1000 iterations. All experiments are conducted on CIFAR-10 with a single NVIDIA-A100 GPU. "-" denotes out-of-memory issue.

|  | IPC | DC | DSA | DM | MTT | IDM | DANCE | **PTLM** |
|---|---|---|---|---|---|---|---|---|
| **Run** | 1 | 0.16 | 0.22 | 0.08 | 0.36 | 0.50 | 0.11 | 0.12 |
| **Time** | 10 | 3.31 | 4.47 | 0.08 | 0.40 | 0.48 | 0.12 | 0.13 |
| **(Sec)** | 50 | 15.74 | 20.13 | 0.08 | - | 0.58 | 0.12 | 0.15 |
| **GPU** | 1 | 3515 | 3513 | 3323 | 2711 | 3223 | 2906 | 3121 |
| **Memory** | 10 | 3621 | 3639 | 3455 | 8049 | 3179 | 3045 | 3371 |
| **(MB)** | 50 | 4527 | 4539 | 3605 | - | 4027 | 3549 | 3894 |

computational time and higher memory usage, making it hard to balance efficiency and effectiveness. Table 6 shows that MTT (Cazenavette et al., 2022), a bi - level optimization method, has good performance but low practical distillation efficiency. DANCE (Zhang et al., 2024), which uses a distribution - matching framework, is highly efficient without sacrificing quality. PTLM is as efficient as DANCE and outperforms it in terms of performance.

# D    ABLATION STUDY

Initially, we evaluated the effectiveness of each module for PTLM individually, the experimental results are presented in Table 7. Then, the sensitivity experimental results for the control factors $\lambda$ and $\tau$ of pseudo-trajectory matching and pseudo-label matching are presented in the appendix.

**Components Evaluation.**    The three main modules—matching capacity theorem (Section 2), pseudo-trajectory matching, and pseudo-label matching (Section 3)—are evaluated, alongside the introduced factoring techniques. As shown in Tables 1 and 2, the matching capacity positively impacts DM-based methods. Notably, the proposed innovations significantly enhance distillation performance individually, with the integrated PTLM framework delivering the greatest gains. This indicates the modules' inherent compatibility and seamless integration. Extensive experimental results validate the effectiveness of the proposed modules, highlighting PTLM's critical role in achieving superior performance across diverse datasets.

Table 7: **Ablation study on three main modules of PTLM.** "✓" denotes the module is included, and "-" ortherwise. "Fac." denotes the Factoring technique. "Cap." denotes the capacity theorem

| Fac. | Cap. | PTM | PLM | CIFAR-10 | | |
|---|---|---|---|---|---|---|
| | | | | 1 | 10 | 50 |
| | ✓ | ✓ | ✓ | $40.4_{\pm0.5}$ | $53.7_{\pm0.6}$ | $72.4_{\pm0.7}$ |
| ✓ | | ✓ | ✓ | $45.7_{\pm0.2}$ | $69.8_{\pm0.5}$ | $75.3_{\pm0.2}$ |
| ✓ | ✓ | | ✓ | $41.4_{\pm0.6}$ | $62.1_{\pm0.1}$ | $71.4_{\pm0.5}$ |
| ✓ | ✓ | ✓ | | $46.8_{\pm0.7}$ | $67.6_{\pm0.4}$ | $73.8_{\pm0.1}$ |
| ✓ | ✓ | ✓ | ✓ | $\mathbf{48.9}_{\pm0.3}$ | $\mathbf{71.3}_{\pm0.3}$ | $\mathbf{78.1}_{\pm0.2}$ |

Table 8: **Ablation on the sensitivity of pseudo-trajectory matching epochs factor $\lambda$.** The evaluation is conducted on CIFAR-10 with 10 images per class.

| $\lambda$ | 1 | 2 | 3 | 4 | 5 | 6 | 7 | 8 | 9 | 10 |
|---|---|---|---|---|---|---|---|---|---|---|
| Acc. | 71.3 | 70.1 | 69.2 | 68.7 | 68.1 | 67.6 | 67.2 | 67.3 | 66.8 | 66.4 |

Table 9: **Ablation on the sensitivity of pseudo-label matching sharpen factor $\tau$.** The evaluation is conducted on CIFAR-10 with 10 images per class.

| $\tau$ | 0.01 | 0.02 | 0.08 | 0.09 | 0.1 | 0.2 | 0.3 | 0.9 | 1.0 |
|---|---|---|---|---|---|---|---|---|---|
| Acc. | 69.5 | 69.8 | 70.4 | 71.1 | 71.3 | 64.2 | 60.5 | 60.1 | 58.7 |

# E   SENSITIVITY ANALYSIS

This section will undertake a detailed analysis of the impact of regulatory factor $\lambda$ in pseudo-trajectory matching and sharpening factor $\tau$ in pseudo-label matching in PTLM.

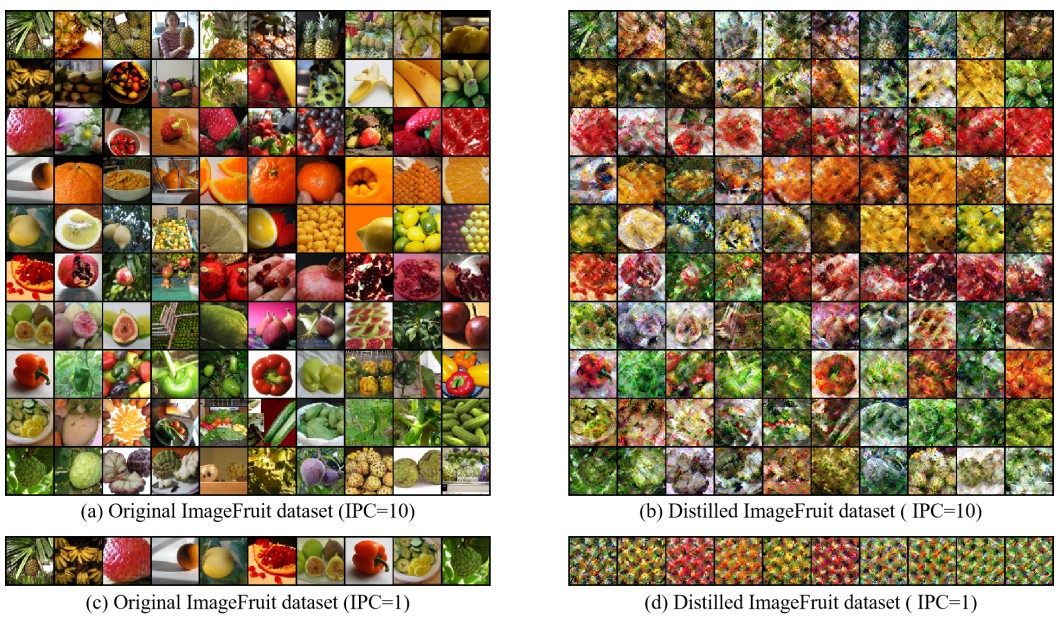

(a) Original ImageFruit dataset (IPC=10)  (b) Distilled ImageFruit dataset ( IPC=10)

(c) Original ImageFruit dataset (IPC=1)  (d) Distilled ImageFruit dataset ( IPC=1)

Figure 2: Example original and distillation images of $128 \times 128$ **ImageFruit** under different IPC setting.

**Sensitivity of PTM Regulatory Factor $\lambda$.** As shown in Table 8, we evaluated the matching impact of the complete pre-trained model training trajectory and partial trajectory. When $\lambda$ is small, the impact on PTLM is not significant. This is because the alterations to the pre-trained model parameters are not sufficiently evident at this subsequent training stage. There will be no significant modifications to the data's characteristic representation form. As $\lambda$ increases, the performance of PTLM declines gradually. This is because the injection of only 10% of the pre-trained parameters into the feature extractor is insufficient to restrict the direction of data feature representation. Consequently, feature transfer continues to impede the distillation performance.

**Sensitivity of PLM Sharpen Factor $\tau$.** The results of the pseudo-label matching experiment are shown in Table 9. These findings reveal that an excessively small or large sharpening factor, $\tau$, negatively affects the performance and reliability of PTLM. Specifically, when $\tau$ is large, the pseudo-labels become overly uniform, leading to misleading and uninformative guidance for the model. Conversely, when $\tau$ is small, the pseudo-labels closely resemble the original labels, failing to capture the correct and diverse distribution of the original dataset. Empirical evidence indicates that PTLM achieves optimal performance when $\tau$ is set to 0.1, striking a balance between diversity and accuracy.

# F    VISUALIZATION

Fig. 2 showcases the visualization results of ImageFruit under diverse IPC settings. When we compare subfigures (a) and (b), it becomes evident that with an IPC value of 10, the images maintain their overall characteristics quite effectively. The colors of these images are largely in line with the target hues of the objects portrayed. Even though the visuals show a degree of abstraction, the object categories can still be recognized. Meanwhile, subfigures (c) and (d) demonstrate that when the IPC is set to 1, the synthesized dataset turns entirely abstract, making it impossible to manually discern specific categories. Nevertheless, the color distribution in these images remains consistent with that of the original ones.

This phenomenon likely results from the varying capacity of the synthesized dataset at different IPC levels. At IPC is 10, the dataset can spread image information from the original one more efficiently, preserving key features and adding details. At IPC = 1, the limited capacity can't hold much original - dataset information, making the images resemble random noise.

