# OpenReview forum: "On Dataset Distillation with Two-fold Pseudo-Distribution Matching"
_ICLR.cc/2026/Conference — Submitted to ICLR 2026_

### Official Review · Reviewer_8HKu · 2025-10-19

**Soundness:** 3
**Presentation:** 2
**Contribution:** 2
**Rating:** 2
**Confidence:** 5

**Summary:**

The paper proposes a new dataset distillation method called Two-fold Pseudo-Distribution Matching (PTLM) to address the challenges of feature and distribution shifts inherent in DM-based approaches. While the experimental results show promising, and in many cases state-of-the-art, performance, the novelty of the core components is somewhat limited.

**Strengths:**

1. The paper introduces and validates a interesting concept, "Matching Capacity," which dictates a strategy for sample-selection-based method: prioritizing typical samples when IPC is low, and incorporating diverse samples as IPC increases.
2. The proposed PTLM method achieves state-of-the-art distillation performance across a wide range of low-, medium-, and high-resolution datasets and various Images Per Class (IPC) settings, notably on CIFAR-100 and TinyImageNet.
3. PTLM is reported to be as efficient as DANCE, which suggests it maintains a competitive balance of efficiency and effectiveness compared to other high-performance but computationally intensive methods.

**Weaknesses:**

1. The two-fold matching components, PTM and PLM, appear to be heavily inspired by or extensions of existing work, particularly DANCE [1] and the general concept of pseudo-labeling in dataset distillation [2,3].
2. The ablation only shows the effect of adding the proposed modules to the naive DM baseline, while a detailed module comparison with DANCE is required, since the design of components is similar to DANCE.
3. Some statements lack evidence. For example, In Line 208,   "random cropping may result in a negative parameter fusion, which in turn causes the extracted features to move from aggregation to a more discrete distribution."  In Line 215, "By simulating the training process, the feature extractor is capable of normalizing the mapping directions of samples with similar characteristics".
4. Missing latest SOTA DM-based methods [4,5,6].

[1] DANCE: Dual-View Distribution Alignment for Dataset Condensation. IJCAI 2024

[2] A Label is Worth a Thousand Images in Dataset Distillation. NeurIPS 2024

[3] Soft-Label Dataset Distillation and Text Dataset Distillation. IJCNN 2021

[4] M3D: Dataset Condensation by Minimizing Maximum Mean Discrepancy. AAAI 2024

[5] Diversified Semantic Distribution Matching for Dataset Distillation. ACM MM 2024

[6] Dataset Distillation with Neural Characteristic Function: A Minmax Perspective. CVPR 2025

**Questions:**

1. There is no experiments that show Maching Capacity and its corresponding application solution is plug-and-play for any DM-based method.

---

### Official Review · Reviewer_uxRa · 2025-10-25

**Soundness:** 3
**Presentation:** 2
**Contribution:** 2
**Rating:** 4
**Confidence:** 4

**Summary:**

This paper introduces a novel Two-fold Pseudo-Distribution Matching (PTLM) approach for dataset distillation, which addresses feature and distribution shifts inherent in the Distribution Matching (DM) paradigm. The authors propose Matching Capacity, a method for selecting typical and diverse samples based on the images-per-class (IPC) of synthetic datasets. PTLM combines Pseudo-Trajectory Matching (PTM) to mitigate feature shifts and Pseudo-Label Matching (PLM) to address distribution shifts. Extensive experiments show that PTLM outperforms state-of-the-art methods while maintaining efficiency.

**Strengths:**

1. The introduction of Matching Capacity enhances the distillation process by adjusting sample selection based on IPC.

2. PTM and PLM effectively reduce feature and distribution shifts, improving distillation quality.

3. The method shows superior performance across multiple datasets with high computational efficiency.

4. The method is a plug-and-play component for existing DM-based distillation methods.

**Weaknesses:**

**Limited Performance Comparison**: Although the authors use DANCE [1] as a baseline, there have been subsequent emerging methods, such as [2] [3]. The authors need to include a broader comparison with these more recent methods.

**Limited Novelty**: The capacity problem proposed by the authors has been pointed out and addressed with different solutions in a number of existing works [4] [5] [6] [7] [8]. The authors should clarify how their method differs from these approaches. Specifically, the loss function used in $L_{PTM}$ is identical to that in DM [9], with the only difference being the feature extractor used. The selection of feature extractors in [1] and [3] provides similar solutions. Additionally, $L_{PLM}$ simply computes cross-entropy between the labels generated by the synthetic dataset and the labels from the real dataset obtained using a pre-trained model, a technique also seen in [10] and [11].

**Confusion with $L_{PLM}$ Design**: I am puzzled by the design of $L_{PLM}$. Since the authors have already extracted soft labels from the real dataset using a pre-trained model, why not use these soft labels and compute the KL divergence with the synthetic dataset's soft labels instead of converting the real dataset's soft labels into hard labels before calculating the cross-entropy loss?

**Incomplete Experiments**: The authors primarily perform experiments on CIFAR and subsets of ImageNet. Although the class matching approach is indeed limited by computational costs and cannot be deployed on ImageNet-1k, I would like the authors to provide related experiments or perform evaluations on more datasets, as suggested in [8]. Furthermore, the authors used ResNet-10 and DenseNet-121 as test architectures in the cross-architecture generalization experiments. Could the authors clarify whether they have conducted tests on more widely used experimental setups such as ResNet-18, AlexNet, and VGG-11?

**Disorganized Structure**: The authors spend excessive space describing the simple issue of capacity and provide overly detailed qualitative explanations of two straightforward submodules, which results in the omission of ablation experiments from the main text. This section needs significant revision.

[1] DANCE: Dual-View Distribution Alignment for Dataset Condensation. In IJCAI, 2024.

[2] Going Beyond Feature Similarity: Effective Dataset Distillation based on Class-aware Conditional Mutual Information. In ICLR, 2025.

[3] Dataset Distillation with Neural Characteristic Function: A Minmax Perspective. In CVPR, 2025.

[4] SelMatch: Effectively Scaling Up Dataset Distillation via Selection-Based Initialization and Partial Updates by Trajectory Matching. In ICML, 2024.

[5] Prioritize Alignment in Dataset Distillation. In arXiv, 2024.

[6] Towards Lossless Dataset Distillation via Difficulty-Aligned Trajectory Matching. In ICLR, 2024.

[7] Curriculum Coarse-to-Fine Selection for High-IPC Dataset Distillation. In CVPR, 2025.

[8] Emphasizing Discriminative Features for Dataset Distillation in Complex Scenarios. In CVPR, 2025.

[9] Dataset Condensation with Distribution Matching. In WACV, 2023.

**Questions:**

Please refer to the weaknesses.

---

### Official Review · Reviewer_ZqLE · 2025-10-27

**Soundness:** 3
**Presentation:** 2
**Contribution:** 2
**Rating:** 2
**Confidence:** 4

**Summary:**

This paper aims to advance Distribution Matching (DM), a category of dataset distillation methods, by introducing three modules: matching capacity, pseudo-trajectory matching, and pseudo-label matching.
The matching capacity module adaptively prunes a portion of real samples based on a pruning factor that linearly increases over training iterations. For IPC (images per class) larger than a threshold alpha, more typical samples are discarded, while for IPC below the threshold, more atypical samples (those far from the feature mean) are removed.
The pseudo-trajectory matching module modifies DM by matching features from a model whose parameters are linearly interpolated between a random initialization and a pretrained model, with the weight toward the pretrained model gradually increasing during training.
The pseudo-label matching module adds a loss term that minimizes the cross-entropy between the average soft-label distributions of the real and synthetic datasets.
Experimental results show that the proposed method, PTLM, achieves comparable or improved performance over prior dataset distillation methods.

**Strengths:**

DM is a computationally efficient distribution-matching approach, though its performance scalability has lagged behind trajectory-based methods (e.g., MTT, Cazenavette et al., 2022) and large-scale methods (e.g., SRe2L, Yin et al., 2023).

This paper proposes three conceptually sound modules to improve DM’s performance scalability across different IPC regimes—(10–50) for low/medium-resolution datasets and (1–10) for high-resolution datasets—and demonstrates noticeable performance gains.

**Weaknesses:**

- Ablation and module interplay: The paper introduces three modules intended to enhance DM under different IPC regimes, but their interactions are not clearly analyzed. The only ablation (Table 7 for CIFAR-10 with IPCs 1-50, Appendix) shows inconsistent trends—some module combinations degrade performance (e.g., the third line in Table 7). The paper should analyze how and why these modules interact differently across datasets (including small, medium, high-resolutions) and IPCs.

- Limited evaluation: Recent distillation works evaluate methods under much larger IPCs—approaching full-dataset performance—to test scalability. This paper evaluates only smaller IPC regimes and uses ConvNets. It would strengthen the claims to include results for larger IPCs and larger networks (e.g., ResNet architectures), especially since DM is computationally scalable method.

- Missing references: Prior works such as DATM (Guo et al., 2023) and SelMatch (Lee et al., 2024) have discussed how the optimal difficulty (atypicality) of synthetic datasets depends on IPC regimes—by matching trajectories of varying feature difficulty or by selecting different difficulty-level of real samples. These are closely related to the “matching capacity” concept here. The authors should cite these works and explicitly highlight how their approach differs.

- Arbitrary design choices: The threshold parameter alpha in Eq. (4) is crucial for deciding whether an IPC is “high” or “low.” However, the choice of alpha likely depends on the dataset size and class count, yet no guidance is provided. Prior works (DATM; SelMatch) offer principled criteria based on IPC levels. Similar justification or sensitivity analysis should be included.

**Questions:**

1. Could the authors provide a more detailed ablation among the three key modules? Additionally, in Table 7, does “factoring technique” refer to the pruning factor? Please clarify.

2. Can the evaluation be extended to larger IPC regimes and larger architectures (e.g., ResNet)?

3. How should one choose the threshold alpha in Eq. (4) distinguishing high vs. low IPC settings across datasets?

4. Please elaborate on how the matching capacity module differs conceptually and algorithmically from DATM (Guo et al., ICLR 2023) and SelMatch (Lee et al., ICML 2024).

---

### Official Review · Reviewer_kQUW · 2025-11-02

**Soundness:** 2
**Presentation:** 3
**Contribution:** 2
**Rating:** 4
**Confidence:** 4

**Summary:**

This paper tackles feature and distribution shifts in Distribution Matching (DM)-based dataset distillation by proposing PTLM (Pseudo-Trajectory matching and Pseudo-Label Matching). The authors introduce the concept of "Matching Capacity," suggesting that low IPC (images-per-class) settings benefit from typical samples while high IPC settings require diverse samples. The PTM component gradually injects pre-trained model parameters to mitigate feature shift, while PLM uses sharpened predictions as pseudo-labels to address distribution shift. Experiments across multiple datasets show improvements over existing DM-based methods, achieving state-of-the-art results on several benchmarks.

**Strengths:**

1. The paper includes extensive experiments across low, medium, and high-resolution datasets, demonstrating consistent improvements. The cross-architecture evaluation (Table 5) effectively shows the transferability of distilled datasets.

2.  The matching capacity mechanism can be integrated into any DM-based method without introducing additional hyperparameters for PTM, making it accessible and practical. The claimed efficiency preservation while improving performance is valuable for real-world applications.

3. The paper provides good intuition for why different matching strategies are needed at different IPC levels. The two-fold approach to addressing feature shift and distribution shift separately is logical and well-motivated through empirical observations.

**Weaknesses:**

1. Limited theoretical justification: The "matching capacity" concept lacks formal analysis or theoretical grounding. Why does the threshold occur around IPC=50? The paper acknowledges "struggle" at IPC 50-100 but doesn't adequately explain this or provide a principled way to determine the threshold adaptively. The claim that this is "plug-and-play" is undermined by the fixed α parameter.

2. Incremental novelty: Both PTM and PLM build heavily on existing ideas—PTM is similar to progressive training/model fusion approaches (even citing DANCE as inspiration), while PLM essentially applies knowledge distillation with label smoothing. The combination is sensible but not particularly innovative. The improvements over DANCE are often modest (0.5-2% in many cases).

3. Incomplete experimental analysis: The paper lacks important ablations—what is the individual contribution of PTM vs. PLM vs. matching capacity? How sensitive is performance to the sharpening factor τ, scaling factor λ, and threshold α? The paper mentions these are in the appendix (sections C, D, E), but core ablations should be in the main paper. Additionally, computational cost comparisons are deferred to the appendix.

4. Questionable design choices: Filtering out incorrectly classified samples from the real dataset (Section 3.2) seems problematic: this could introduce bias and doesn't account for hard-to-classify but valid samples. The linear injection schedule in Eq. 5 is not justified over alternatives. The distinction between "typical" and "diverse" samples based on cosine similarity to the mean is overly simplistic.

5. Writing and clarity issues: Some claims are overclaimed (e.g., "top AI researcher" framing in methodology). The related work section is missing, making it hard to position contributions. Mathematical notation could be cleaner (e.g., f(θ,·) notation is awkward). The paper could better discuss limitations and failure cases.

**Questions:**

1. **Adaptive threshold selection**: The authors mention "future research directions will encompass the investigation of adaptive IPC judgment" when discussing the struggle at IPC 50-100. Could you provide more insight into why the fixed threshold α=50 was chosen, and have you explored any adaptive mechanisms? Can you characterize when matching capacity helps vs. hurts performance?

2. **Component contributions**: What are the individual contributions of pseudo-trajectory matching, pseudo-label matching, and matching capacity to the overall performance? An ablation study showing PTM-only, PLM-only, and matching capacity-only results would clarify which components drive the improvements and whether they provide complementary benefits.

---

### Meta-Review · Area_Chair_m73q · 2026-01-01

**Summary:**

This paper tackles feature and distribution shifts in Distribution Matching (DM)-based dataset distillation by proposing PTLM (Pseudo-Trajectory matching and Pseudo-Label Matching).
The main concerns are incremental novelty, limited theoretical justification, incomplete experimental analysis.
No rebuttal is provided.  All reviewers would not change their score. So this paper should be rejected.

**Reviewer Concerns:**

No rebuttal is provided. The main concerns are incremental novelty, limited theoretical justification, incomplete experimental analysis. These concerns are still outstanding.

**Reviewer Scores:**

No rebuttal is provided.
Reviewer kQUW would not change their score.
Reviewer ZqLE would not change their score.
Reviewer uxRa would not change their score.
Reviewer 8HKu would not change their score.

---

### Decision · Program_Chairs · 2026-01-26

Reject